# Link between supercurrent diode and anomalous Josephson effect revealed by gate-controlled interferometry

S. Reinhardt [1], T. Ascherl [1], A. Costa [2], J. Berger[1], S. Gronin[3], G. C. Gardner[3], T. Lindemann[3,4], M. J. Manfra [3,4,5,6], J. Fabian [2], D. Kochan [2,7,8], C. Strunk [1] & N. Paradiso [1]

In Josephson diodes the asymmetry between positive and negative current branch of the current-phase relation leads to a polarity-dependent critical current and Josephson inductance. The supercurrent nonreciprocity can be described as a consequence of the anomalous Josephson effect —a $\varphi_0$-shift of the current-phase relation— in multichannel ballistic junctions with strong spin-orbit interaction. In this work, we simultaneously investigate $\varphi_0$-shift and supercurrent diode efficiency on the same Josephson junction by means of a superconducting quantum interferometer. By electrostatic gating, we reveal a direct link between $\varphi_0$-shift and diode effect. Our findings show that spin-orbit interaction in combination with a Zeeman field plays an important role in determining the magnetochiral anisotropy and the supercurrent diode effect.

In solids, spin-orbit interaction (SOI) makes it possible to control orbital degrees of freedom by acting on the electron spin, and vice-versa[1,2]. In superconductors[3], the impact of SOI can be particularly spectacular, since it enables phenomena which go beyond the realm of conventional $s$-wave superconductors, as e.g. topological phases[4], finite-momentum superconductivity[5–7], Lifshitz invariant[8–10], Ising superconductivity[11], anomalous Josephson effect[12–16], and intrinsic supercurrent diode effect[17–31]. In what follows, we shall focus on the last two effects and on their relation in Josephson junctions.

The anomalous Josephson effect manifests itself in a phase offset $\varphi_0$ at zero current, $I(\varphi_0) = 0$, in the current-phase relation (CPR)[15,16,32–36]. This also implies a finite supercurrent at zero phase difference $I(\varphi = 0) \neq 0$. The effect requires the simultaneous breaking of both inversion and time-reversal symmetry[37], which can be provided by SOI and Zeeman field, respectively.

The same symmetries need to be broken in order to observe the supercurrent diode effect (SDE), namely, the dependence of the critical current on the bias polarity. This effect can be trivially obtained, e.g., in asymmetric superconducting quantum interference devices (SQUIDs)[38] or, more generally, when in a film an inhomogeneous supercurrent distribution is coupled to a flux. Recently, it was shown[17–19,22–27,30,31] that supercurrent rectification can as well emerge as an intrinsic feature of *homogeneous* quasi-2D systems subjected to a Zeeman field. Such nontrivial SDE is a new precious probe of the condensate physics (in films)[10] and of Andreev bound states (ABSs) in Josephson junctions[27,30], including possible topological properties[39–41].

Several mechanisms have been proposed to explain such intrinsic SDE in films[10,42–46] and Josephson junctions[47–53]. In superconducting-normal-superconducting (SNS) junctions the supercurrent can be computed in terms of the ABSs in the N weak link. In experiments, ABSs can be directly probed by tunnel spectroscopy[36,54,55], while their effect on the CPR can be deduced from IV-characteristics, inductance versus current measurements[18,27,56] and SQUID experiments[57].

[1]Institut für Experimentelle und Angewandte Physik, University of Regensburg, Regensburg, Germany. [2]Institut für Theoretische Physik, University of Regensburg, Regensburg, Germany. [3]Birck Nanotechnology Center, Purdue University, West Lafayette, IN, USA. [4]Department of Physics and Astronomy, Purdue University, West Lafayette, IN, USA. [5]School of Materials Engineering, Purdue University, West Lafayette, IN, USA. [6]Elmore Family School of Electrical and Computer Engineering, Purdue University, West Lafayette, IN, USA. [7]Institute of Physics, Slovak Academy of Sciences, Bratislava, Slovakia. [8]Center for Quantum Frontiers of Research and Technology (QFort), National Cheng Kung University, Tainan, Taiwan. ✉e-mail: nicola.paradiso@physik.uni-regensburg.de

Early observations on Josephson diodes were interpreted in terms of $\varphi_0$-shift in ballistic systems with skewed CPR[18,22–24,56]. Within this picture, the SDE ultimately originates (as it does the $\varphi_0$-shift) from SOI. An alternative model, proposed by Banerjee et al.[30] based on the theory of ref. 50, explains the same effect in terms of a purely orbital mechanism. To date, it is not clear yet to which extent the two mechanisms (namely, the SOI-based and the purely orbital mechanism) contribute to the supercurrent rectification observed in experiments.

In this work, we make use of an asymmetric SQUID with mutually orthogonal junctions to directly measure both the anomalous $\varphi_0$-shift and the SDE on the same junction. By gating, we can electrostatically control both effects and highlight their relation. Finally, by measuring the temperature dependence of the $\varphi_0$-shift and of the diode efficiency we highlight the role of the higher harmonics of the CPR for the emergence of the SDE. We comment on our results in light of alternative models proposed in the literature and compare the temperature dependence of $\varphi_0$ and the SDE to the predictions of a minimal theoretical model.

Figure 1a shows a scheme of our SQUID. The device is fabricated starting from a molecular beam epitaxy-grown heterostructure featuring an InGaAs/InAs/InGaAs quantum well, capped by a 5-nm-thick epitaxial Al film[18,58–60]. The quantum well hosts a 2D electron gas (2DEG) with a proximity-induced superconducting gap inherited from the Al film. By deep wet etching, we define an asymmetric SQUID loop. The actual geometry is shown in the false-color scanning electron microscopy image in Fig. 1b, where the turquoise areas indicate the pristine Al/InGaAs/InAs/InGaAs regions, while the gray areas refer to deeply etched (insulating) regions. To obtain the two normal (N) weak links, we selectively etch the Al film (yellow areas in Fig. 1b). The reference Josephson junction 1 (JJ1) is 28-µm-wide and 120-nm-long, whereas the

Josephson junction 2 (JJ2) is 2.7-µm-wide and 100-nm-long. Finally, a gate is fabricated on top of JJ2, which allows us to control the electron density in the N-link and thus the critical current $I_{c,2}$ of this junction. The two junctions are mutually perpendicular, so that an in-plane magnetic field $\vec{B}_{ip}$ parallel to the short junction, i.e., along $\hat{y}$, (Fig. 1a, b) will induce magnetochiral effects[18] in the short junction only, and not in the reference junction. Here, we take as positive $\hat{z}$ direction that perpendicular to the 2DEG and directed from the substrate towards the Al (Fig. 1a), which corresponds to the direction opposite to the built-in electric field in the quantum well[18,61] which provides SOI in the 2DEG[18,61].

We measure differential resistance in a 4-terminal geometry as a function of DC current. In what follows, we indicate as $I_c$ the (measured) SQUID critical current (see Methods) and as $I_{c,i}$ ($i = 1, 2$) the (deduced) critical current in junction $i$. When needed, we use the superscript + (−) to indicate positive (negative) current from source to drain (Fig. 1a), i.e., supercurrent in the positive (negative) $x$ direction in JJ2. For the in-plane field, we interchangeably use either Cartesian components, or magnitude and angle parametrization, i.e. $\vec{B}_{ip} = B_x\hat{x} + B_y\hat{y} = B_{ip}(\cos\theta, \sin\theta)$, see cartesian axes in Fig. 1a.

## Results

Figure 1c shows the color plot of the differential resistance versus out-of-plane field $B_z$ and DC current $I$, measured with an applied field $B_y = -100$ mT at $T = 40$ mK. The fast oscillations have period 1.8 µT, corresponding to a flux quantum $\Phi_0 = h/2e$ applied to the loop. Such oscillations are superimposed to the Fraunhofer pattern of the reference junction, whose central and first side lobes are visible. The plot displays an evident asymmetry around the $B_z = 0$ and $I = 0$ axes, while it is approximately point-inversion-symmetric around the origin, namely

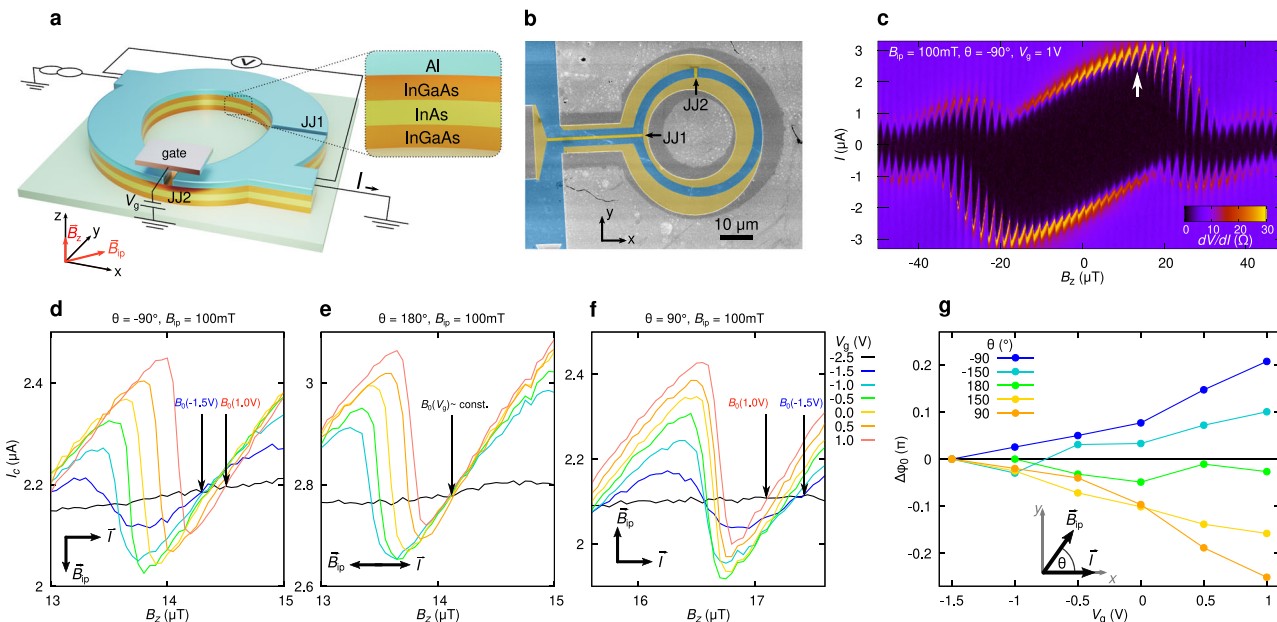

**Fig. 1 | Asymmetric SQUID device with reference junction and gate-controllable $\varphi_0$ junction. a** (Left) Schematic illustration of the device. The SQUID consists of a large reference junction (JJ1) and of a small junction (JJ2) which is coupled to a gate. By applying a magnetic field along $\hat{y}$, a $\varphi_0$-shift is induced in the CPR of the latter junction, which can be controlled by electrostatic gating. (Right) Scheme of the topmost layers of the heterostructure. The black arrow indicates the direction of the positive current bias $I$. **b** False-color scanning electron microscopy image of the device taken before gate patterning. The pristine superconducting Al/InAs leads are highlighted in turquoise, the areas where Al is selectively etched in yellow (including the weak links, highlighted by the black arrows). The remaining parts in gray correspond to deeply etched regions,

where both the Al film and the topmost semiconducting layers are etched. **c** The color plot shows the SQUID differential resistance versus out-of-plane field $B_z$ and current $I$, for $\theta = -90°$ and $B_{ip} = 100$ mT (i.e., $B_x = 0$, $B_y = -100$ mT) at $T = 40$ mK. The white arrow indicates where anomalous phase shifts were measured (see panels **d**–**f**). **d** SQUID critical current $I_c$ as a function of $B_z$, for $\theta$ and $B_{ip}$ as in **c**. The different curves refer to different gate voltages $V_g$. We define $B_0(V_g)$ as the crossing of each curve with the $V_g = -2.5$ V reference curve (black). **e**, **f** Corresponding measurements for the same $B_{ip} = 100$ mT but, respectively, $\theta = 180°$ and $\theta = 90°$. **g** Plot of $\Delta\varphi_0(V_g) \equiv 2\pi A[B_0(V_g) - B_0(V_g = -1.5 \text{ V})]/\Phi_0$, for different $\vec{B}_{ip}$ orientations, i.e., for different $\theta$. Here, $A$ is the loop area and $\Phi_0$ the flux quantum.

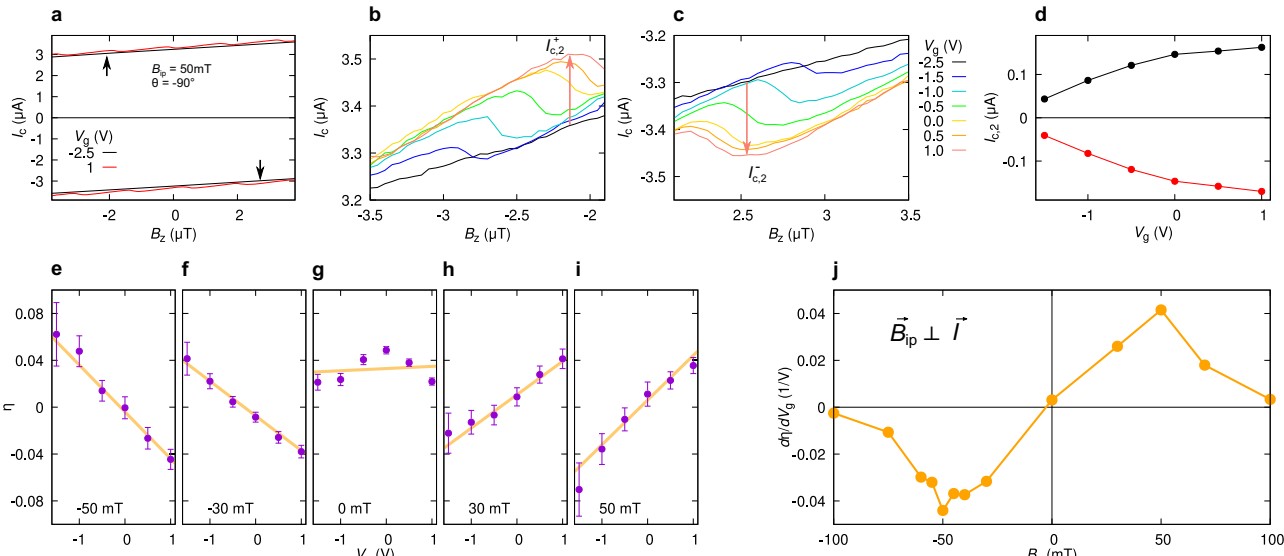

**Fig. 2 | Gate control of the supercurrent diode effect. a** Positive and negative SQUID critical current for an out-of-plane field $B_z$ close to zero, $T = 40$ mK, and $B_y = -50$ mT. The red (black) curve refers to a gate voltage $V_g = 1.0$ V ($V_g = -2.5$ V). The upper and lower arrow indicates the oscillation highlighted in panel **b** and **c**, respectively. They are located asymmetrically in $B_z$, eliminating the trivial diode effect of the background. **b** Positive critical current $I_c^+$ versus out-of-plane field $B_z$. The red arrow indicates $I_{c,2}^+$ for the $V_g = 1$ V-curve, see text. **c** Negative critical current $I_c^-$ versus $B_z$. The red arrow indicates $I_{c,2}^+$ for the $V_g = 1$ V-curve. The different curves in **b** and **c** correspond to different values of the gate voltage $V_g$. Measurements are performed at $T = 40$ mK and $B_y = -50$ mT. **d** $I_{c,2}^+$ and $I_{c,2}^-$ versus $V_g$ for $B_y = -50$ mT. **e-i** Diode efficiency $\eta \equiv 2(I_c^+ - |I_c^-|)/(I_c^+ + |I_c^-|)$ versus $V_g$, for $B_y = -50$ mT (**e**), $-30$ mT (**f**), 0 mT (**g**), 30 mT (**h**), 50 mT (**i**). **j** Slope of the diode efficiency $d\eta/dV_g$, plotted versus $B_y$.

$I_c(B_z) \approx -I_c(-B_z)$. The dominant extrinsic SDE of the SQUID as a whole (since $I_c(B_z) \neq -I_c(B_z)$) is a long-known consequence of screening in the loop, which must be kept distinct from the intrinsic SDE in single homogeneous junctions, which is the effect we are studying. As discussed in the Supplementary Information, screening in SQUIDs arises when there are current-dependent corrections to the fundamental SQUID relation

$$\gamma_1 - \gamma_2 = \frac{2\pi}{\Phi_0} \Phi, \qquad (1)$$

where $\gamma_i$ it the gauge-invariant phase drop at the $i$-th junction, $\Phi_0$ is the flux quantum and $\Phi$ the flux through the loop. The corrections originate either from the additional flux induced by the SQUID current itself, or by the phase drop accumulated along the loop arms[38,62]. The former correction is proportional to the geometric loop inductance, while the latter is proportional to the kinetic inductance of the SQUID arms[63]. As discussed in the Methods, in our sample, screening is mostly dominated by the large kinetic inductance of the thin Al film, whose sheet inductance is $L_\square \sim 30$ pH. The geometric loop inductance is comparatively small, see Supplementary Information. As shall be discussed below, screening effects hinder the measurement of both the absolute value of the anomalous shift $\varphi_0$ and that of the diode efficiency. Indeed, the determination of the absolute $\varphi_0$ is challenging even in SQUID devices with low screening and reference devices[32,36]. One of the difficulties is the fact that an accurate determination of $\varphi_0$ requires a reproducible $B_z$ control on the microtesla scale, while an in-plane field of the order of tens or hundreds of millitesla is swept.

The determination of the *absolute* $\varphi_0$ is usually difficult in systems with large kinetic inductance. Similar to refs. 33,34,36, we shall measure the *relative* shift of the CPR with respect to that for large negative gate voltage. Figure 1d shows SQUID oscillations measured for different gate voltages $V_g$. The measurement is performed with an in-plane field $B_{ip} = 100$ mT applied perpendicular to JJ2 ($\theta = -90°$) at $T = 40$ mK. The oscillations are taken near the maximum of the tilted Fraunhofer pattern, indicated by the upper arrow in Fig. 1c. The black curve

($V_g = -2.5$ V) refers to a completely pinched-off JJ2 ($I_{c,2} = 0$), and serves as zero-current baseline. The first curve for which oscillations are clearly visible is that for $V_g = -1.5$ V. We shall label as $B_0(V_g)$ the crossing with positive slope of each $I_c(V_g)$ curve with the baseline $I_c(V_g = -2.5$ V), see Fig. 1d. As discussed in the Supplementary Information, at these crossings the supercurrent $I_2$ in JJ2 vanishes, therefore its gauge invariant phase difference is, by definition, the anomalous shift $\varphi_0$. We consider variations of $\varphi_0$ with respect to the reference voltage $V_g = -1.5$ V[36], namely, $\Delta\varphi_0 \equiv 2\pi A_{loop}[B_0(V_g) - B_0(V_g = -1.5$ V)]$/\Phi_0$, where $A_{loop} = 1150$ μm² is the loop area.

Figure 1e and f show the results of the same measurements after two subsequent -90° sample rotations, namely, for $\theta = 180°$ and $\theta = 90°$, respectively. From Fig. 1e we deduce that for $\vec{B}_{ip} \parallel \vec{I}$, the curves cross the baseline with positive slope nearly at the same $B_z$ (i.e., $B_0(V_g)$ is constant). Instead, for $\vec{B}_{ip} \perp \vec{I}$, $B_0(V_g)$ monotonically increases (decreases) with $V_g$ for negative (positive) sign of $\hat{e}_z \cdot (\vec{B}_{ip} \times \vec{I})$. This is a clear signature of the magnetochiral nature of the anomalous Josephson effect[18].

The variation of $\Delta\varphi_0(V_g)$ for different $\vec{B}_{ip}$ orientations (i.e., for different $\theta$ with $|\vec{B}_{ip}| = 100$ mT) is plotted in Fig. 1g. We stress that, since we subtract $\varphi_0(V_g = -1.5$ V) (as in the definition of $\Delta\varphi_0$), what is important in Fig. 1g is the monotonic increase or decrease of $\Delta\varphi_0$ with the gate voltage. The graph clearly shows the proportionality of $\Delta\varphi_0$ to[18] $-\hat{e}_z \cdot (\vec{B}_{ip} \times \vec{I})$, as expected by SOI-based models for the anomalous Josephson effect[12]. A graph of $\Delta\varphi_0$ versus the angle $\theta$ is shown in Fig. S4 of the Supplemental Information. As expected, we find the sinusoidal dependence $\Delta\varphi_0 \sim -\sin(\theta)$. For all curves, the magnitude of $|\Delta\varphi_0|$ increases with $V_g$. The monotonic increase is expected, since in this type of InGaAs/InAs/InGaAs quantum wells, a positive gate voltage increases the built-in electric field[61] responsible for the Rashba SOI[18,61].

The main goal of our experiments is to establish a relation between the anomalous $\varphi_0$-shift and intrinsic SDE by measuring both phenomena on the same junction. For this purpose, we investigate SQUID oscillations for both current bias polarities in order to deduce both the positive ($I_{c,2}^+$) and the negative ($I_{c,2}^-$) critical current of JJ2. Figure 2a shows the SQUID interference pattern measured in the

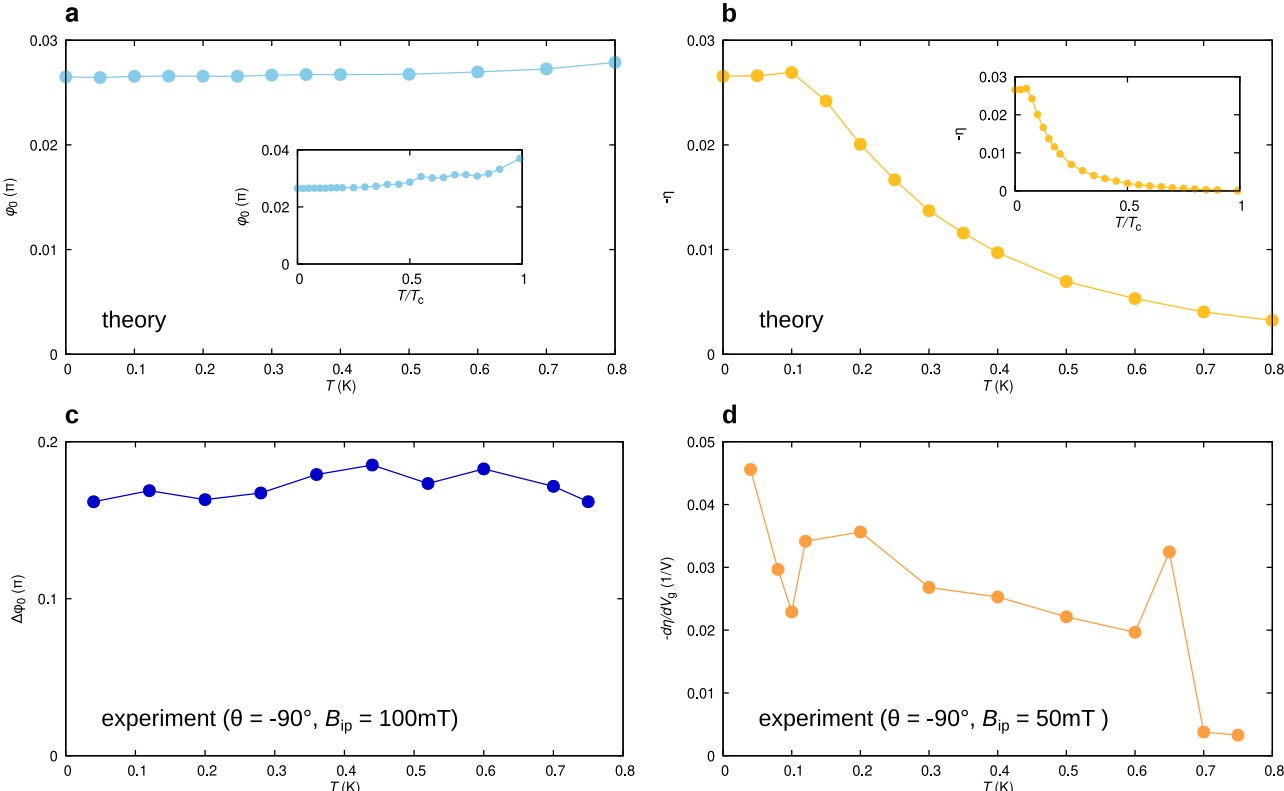

**Fig. 3 | Temperature dependence of $\varphi_0$ and $\eta$: theory and experiment.**
**a** Computed temperature dependence of the (sign changed) anomalous phase shift $\varphi_0$ for a Zeeman parameter of $\lambda_Z = -0.40$ (which would correspond to $B_y = -100$ mT for g-factor $|g| = 12$). We assume $T_c = 2.0$ K as in the experiments. Inset: full-range $\varphi_0(T)$ graph plotted for $T$ up to $T = T_c$. **b** Computed temperature dependence of the supercurrent diode rectification efficiency $-\eta$ for the same

parameters as in **a**. Inset: full-range $-\eta(T)$ graph up to $T = T_c$. **c** Measured $\Delta\varphi_0$ (as defined in the text) at $V_g = 1.0$ V and $B_y = -100$ mT, plotted as a function of temperature. **d** Temperature dependence of $d\varphi_0/dV_g$ (as defined in the text) at $B_y = 50$ mT. For ease of comparison, all main graphs are plotted up to $T = 0.8$ K, the highest temperature for which $\Delta\varphi_0$ and $d\varphi_0/dV_g$ was measurable.

vicinity of $B_z = 0$ for $B_y = -50$ mT at $V_g = 1.0$ V (red), together with the reference curve at $V_g = -2.5$ V (black, where $I_{c,2} = 0$ and SQUID oscillations vanish). Figure 2b shows several $I_c^+(B_z)$ curves for different gate voltages $V_g$, measured at $B_y = -50$ mT and $T = 40$ mK, as in Fig. 2a. We focus on one particular oscillation highlighted by the upper arrow in Fig. 2a. We stress that, owing to screening, one cannot extract the negative critical current in the CPR from the relative minima of $I_c^+(B_z)$. As discussed in the Supplementary Information, it is instead necessary to look at the maxima of $|I_c^-(-B_z)|$. The corresponding $I_c^-$ curves for the opposite $B_z$ range (lower arrow in Fig. 2a) are shown in panel **c**. We use the baseline curve in Fig. 2a (black, $V_g = -2.5$ V) as a reference to extract $I_{c,2}^+$ from data in Fig. 2b and $I_{c,2}^-$ from Fig. 2c. For each $V_g$, $I_{c,2}^+$ corresponds to the maximum in $B_z$ (see red arrow in Fig. 2b) of the difference $I_c^+(B_z, V_g) - I_c^+(B_z, V_g = -2.5$ V). $I_{c,2}^-$ is deduced in a similar way from $I_c^-(B_z, V_g) - I_c^-(B_z, V_g = -2.5$ V).

The resulting $I_{c,2}^+(V_g)$ and $I_{c,2}^-(V_g)$ are plotted in Fig. 2d. As a figure of merit for the supercurrent rectification, we use the supercurrent diode efficiency $\eta \equiv (I_{c,2}^+ - |I_{c,2}^-|)/\langle I_{c,2}\rangle$, with $\langle I_{c,2}\rangle \equiv (I_{c,2}^+ + |I_{c,2}^-|)/2$. The efficiency $\eta(V_g)$ is plotted in Fig. 2e–i for different values of $B_y$. As explained in the Supplementary Information, $\eta$ is subjected to a spurious vertical shift owing to SQUID screening effects. On the other hand, its slope as a function of $V_g$, i.e. $\partial\eta/\partial V_g$ remains unaffected by screening. Our crucial assumption is that, seen as functions of $\vec{B}$ or $T$, $\eta$ and $\partial\eta/\partial V_g$ are proportional, i.e., the larger $\eta$, the larger is its modulation of as a function of $V_g$. Thus, we shall use $\partial\eta/\partial V_g$ as a convenient figure of merit for the magnetochiral effect when considering field or temperature dependence.

Previous reports on Josephson diodes[18,22,24,25] showed that the rectification efficiency is linear in $B_y$, but only up to a certain threshold

(here at $|B_y| = 50$ mT): after that, a clear suppression is observed. Our data in Fig. 2j, deduced from data in Fig. 2e–i, show precisely the same behavior. The suppression at higher field was interpreted in ref. 27 as a signature of a 0-$\pi$-like transition, occurring when two minima in the Josephson energy-phase relation become degenerate.

The results shown so far provide strong evidence of the link between anomalous phase shift and SDE. Both effects are linear in $B_y$. Both $\Delta\varphi_0$ and $\eta$ can be modulated by a gate voltage in the accessible range $V_g \in [-1.5$ V, $1.0$ V]. This is consistent with the SOI-based model[10,18,24,53,56] of the SDE, where the supercurrent rectification is accompanied by the anomalous Josephson shift $\varphi_0$. However, the $\varphi_0$-shift is, per se, not a sufficient condition for the supercurrent non-reciprocity. To break the symmetry between positive and negative part of the CPR, multiple ballistic channels are needed[14,27]. In finite-width junctions with parabolic dispersion, each $i$-channel has a different $\varphi_{0,i}$-shift, owing to the different Fermi velocity $v_{F,i}$: in fact, for a single ballistic channel, $\varphi_{0,i} \propto v_{F,i}^{-2}$[12]. The total CPR is the sum of all single-channel CPR contributions $I_i(\varphi + \varphi_{0,i})$. The sum of skewed CPRs with different $\varphi_{0,i}$-shift leads to an asymmetric total CPR, whose positive and negative branch are different. The skewness (i.e., the content of higher harmonics) of the individual CPRs is crucial, since otherwise the sum of sinusoidal CPRs would always lead to a sinusoidal –i.e., reciprocal– CPR. The relation between $\eta$ and $\varphi_0$ is evident from the comparison between Fig. 1g and Fig. 2e–i. On the other hand, since the SDE also relies on the presence of higher harmonics in the CPR[18], we expect that $\eta$ (and thus $\partial\eta/\partial V_g$) will be highly sensitive to the junction transparency and temperature, as opposed to the anomalous shift $\varphi_0$.

Figure 3a and b respectively show the temperature dependence of $\varphi_0$ and $\eta$ as obtained from a simple theoretical model (described in

Methods and in refs. [27,53]) for a Josephson junction in the short-ballistic limit with Zeeman interaction. The junction separates two semi-infinite superconducting 2DEGs with zero-temperature superconducting gap $\Delta^*(0) \approx 130\,\mu eV$ [60], and critical temperature $T_c \approx 2\,K$. We note that the anomalous shift $\varphi_0$ is nearly $T$-independent, whereas the supercurrent rectification is strongly suppressed already for $T > 100\,mK$.

In Fig. 3c and d we show the measured temperature dependence of $\Delta\varphi_0$ and that of $\partial\eta/\partial V_g$, which we take as measures of $\varphi_0(T)$ and $\eta(T)$, respectively. $\Delta\varphi_0(T)$ is measured at $B_y = -100\,mT$, $(B_{ip} = 100\,mT, \theta = -90°)$, where the field magnitude is set large enough to limit screening effects to an acceptable level. Instead, $\partial\eta/\partial V_g(T)$ is measured at $B_y = 50\,mT$, where the SDE is maximal, see Fig. 2j. Figure 3c shows that $\Delta\varphi_0$ is temperature-independent within the experimental accuracy. In contrast, the (gate modulation of the) supercurrent rectification is clearly suppressed already at temperatures well below $T_c$, as shown in Fig. 3d. Both observations match the corresponding theory predictions.

A comment is in order about the sign and magnitude of the effects. Both our experimental data and analytical model show that if the Rashba SOI-inducing electric field is directed along $-\hat{z}$ (as in ref. [61] and in the Supplementary Information of ref. [18]), $\vec{B}_{ip}$ along $+\hat{y}$, and the positive current bias along $+\hat{x}$, then $\varphi_0 < 0$ [where $I(\varphi_0) = 0$, $\partial_\varphi I(\varphi_0) > 0$] and $\eta > 0$. Instead, the magnitude of $\varphi_0$ and $\eta$ predicted by ballistic theory [12] is smaller than the one measured in our and in other experiments in the literature [32,33]. A possible explanation for this discrepancy could be disorder in and near the junction [32], since diffusive models predict a much larger $\varphi_0$. Another possibility is the enhancement of SOI due to interaction of quantum well electrons with the image charges that are formed in a nearby Al gate. The nontrivial property of the induced image-potential that depends on the electron density of 2DEG gives a feedback on SOI that superimposes with the innate Rashba SOI of the quantum well without metallic gate as demonstrated in refs. [64–66].

## Discussion

The main goal of our study is to elucidate the physical mechanism behind the intrinsic SDE in single, homogeneous Josephson junctions. The effect has been so far explained by two different models: one [14,18,56] is based on the combination of Rashba SOI plus Zeeman interaction (due to an external in-plane field or exchange interaction); the other is a purely orbital mechanism [30,50] based on the finite Cooper pair momentum induced in the superconducting leads by the flux associated to the in-plane field [30]. This flux is finite if the parent superconducting film and the 2DEG are spatially separated.

The main difference between the two pictures is the expected dependence on $V_g$. Such dependence naturally emerges since $V_g$ affects the band alignment and thus the Rashba coefficient $\alpha_R$. Both $\alpha_R$ and the electron density $n$ critically affect $\varphi_0$, which, in turn, determines $\eta$ in multichannel systems. In contrast, the orbital mechanism [30] hardly depends on the gate voltage (Max Geier, Karsten Flensberg, private communication). As discussed in the Supplementary Information, the gate voltage also affects the magnetochiral anisotropy for the inductance [18], an effect that is strictly related to the supercurrent rectification [27]. The observed strong gate dependence of $\varphi_0$ and $\eta$ indicates that the Rashba-based mechanism must certainly play an important role in the SDE. On the other hand, the orbital mechanism cannot be ruled out by our observations: it could still coexist with the spin-orbit-based mechanism.

Finally, we would like to stress that, even though we make use of a SQUID to link $\eta$ to $\varphi_0$, our point does not concern the (trivial and long known [38]) SDE of the asymmetric SQUID as a whole. Our focus is exclusively on the intrinsic SDE in a single, homogeneous junction (JJ2).

In conclusion, we have shown the coexistence of anomalous Josephson effect and supercurrent rectification by measuring both effects on the same Josephson junction embedded in a SQUID. The observed gate voltage and temperature dependence are compatible with a spin-orbit based picture where supercurrent rectification arises in multichannel junctions with anomalous shift $\varphi_0$ and skewed current-phase relation.

Josephson diodes based on $\varphi_0$-junctions are important for both fundamental research and applications, e.g. as sensors for readout of racetrack memory devices [67,68]. They are novel and powerful probes of symmetry breaking in 2D superconductors [21,49,69] and possible probes of topological phase transitions [39]. A recent proposal [70] suggested that the anomalous Josephson effect might be used in multiterminal junctions to obtain compact nonreciprocal devices as, e.g., circulators for rf-applications [71].

## Methods

### Experimental methods

The heterostructure is grown by molecular beam epitaxy. The full layer sequence is reported in the Supplementary Information. The most relevant layers are the topmost ones, namely, the nominally 5-nm-thick Al film at the sample surface, a 10-nm-thick $In_{0.75}Ga_{0.25}As$ layer acting as a barrier, a 7-nm InAs layer hosting the 2DEG, followed by another $In_{0.75}Ga_{0.25}As$ barrier of thickness 4 nm. Structures are defined by electron beam lithography. The selective etching of Al is performed by wet chemical etching using Transene D. Deep etching processes (where the 2DEG is removed altogether) is performed using a phosphoric acid-based solution.

Transport measurements are performed in a 4-point configuration using standard lock-in techniques. To determine the SQUID critical current $I_c^+$ and $I_c^-$ as defined in the text we take $dV/dI = 6\,\Omega$ as a threshold.

To determine the correct offset for $B_z$ we look at the symmetry of the plot of $R = dV/dI$ versus $I$ and $B_z$, see e.g., Fig. 1c. Since $R(I, B_z) \approx R(-I, -B_z)$, the center of inversion symmetry of the plot allows us to determine the applied out-of-plane field which corresponds to an effective $B_z = 0$.

### Circuit model

A circuit model of the asymmetric SQUID device, which includes the effect of inductive screening in the loop and the reference junction is described in the Supplementary Information. The model can reproduce the experimentally found $I_c(B_z)$ curves when using realistic parameters for the kinetic inductance of the aluminum electrodes. A detailed discussion of the model and simulation results are provided in the Supplemental Information. The most important findings from the simulations regarding our data evaluation are:

- With our device parameters, inductive screening effects do not affect the extraction of $\varphi_0$. When $\varphi_0 = 0$, we find that all curves intersect the reference line ($I_{c,2} = 0$) in the same point.
- The critical current $I_{c,2}$ extracted from the SQUID oscillations is proportional to the actual critical current of JJ2. In general, the proportionality constant can be slightly different for the positive and negative critical current. For the evaluation of $\eta(V_g)$ this means that we can only analyze $d\eta/dV_g$, as $\eta(V_g)$ will have an uncontrolled offset.

### Theoretical methods

Our theoretical model, initially developed in refs. [27,53], describes the experimentally relevant system in terms of a short S−N−S Josephson junction that couples two semi-infinite $s$-wave superconducting (S) regions with inherently strong Rashba SOI through a thin delta-like normal-conducting (N) link. Nontrivial solutions of the 2D

Bogoliubov–de Gennes equation[72]

$$\begin{bmatrix} \hat{\mathcal{H}} & \hat{\Delta}(x) \\ \hat{\Delta}^{\dagger}(x) & -\hat{\sigma}_y(\hat{\mathcal{H}})^{*}\hat{\sigma}_y \end{bmatrix}\Psi(x,y) = E\Psi(x,y), \qquad (2)$$

with the single-electron Hamiltonian

$$\hat{\mathcal{H}} = \left[ -\frac{\hbar^2}{2m}\left(\frac{\partial^2}{\partial x^2}+\frac{\partial^2}{\partial y^2}\right) - \mu \right]\hat{\sigma}_0 + \alpha_R\left(k_y\hat{\sigma}_x - k_x\hat{\sigma}_y\right) \\ + \left(V_0\hat{\sigma}_0 + V_Z\hat{\sigma}_y\right)d\delta(x), \qquad (3)$$

determine the energies $E$ and wave functions $\Psi(x,y)$ of the Andreev bound states[73], which are at the heart of the coherent Cooper-pair supercurrent transport along the $\hat{x}$-direction; $\hat{\Delta}(x)$ corresponds to the $s$-wave superconducting pairing potential that we approximate by

$$\hat{\Delta}(x) = \Delta^{*}(T)\left[\Theta(-x) + e^{i\varphi}\Theta(x)\right], \qquad (4)$$

where $\Delta^{*}(T) = \Delta^{*}(0)\tanh(1.74\sqrt{T_c/T - 1})$ approximates the temperature-dependent proximity-induced superconducting gap [from the experimental data, the induced gap at zero temperature was estimated as $\Delta^{*}(0) \approx 130\,\mu eV$ and the critical temperature as $T_c \approx 2\,K$] and $\varphi$ indicates the phase difference between the two superconducting regions. The Rashba SOI that is present throughout the whole system is parameterized by $\alpha_R$, $V_0$ and $V_Z$ represent the scalar (spin-independent) and Zeeman (spin-dependent) potentials inside the delta-like N link of thickness $d$−the magnetic field causing the Zeeman splitting is thereby aligned perpendicular to the current direction (i.e., along $\hat{y}$)−, $\mu$ is the Fermi energy, $m$ the (effective) quasiparticle mass, and $\hat{\sigma}_0$ and $\hat{\sigma}_i$ refer to the $2 \times 2$ identity and $i$th Pauli spin matrix, respectively.

After determining the Andreev-state energies $E_n(\varphi)$ (with $n$ labeling the $n$-th tranverse channel) as a function of the superconducting phase difference $\varphi$, we apply the quantum-mechanical current operator to the corresponding bound-state wave functions inside the N link to compute in the first step the Josephson CPRs $I(\varphi)$ and obtain in the second step the direction-dependent critical currents necessary to quantify the SDE. In the simultaneous presence of SOI and Zeeman interaction, the bound-state energies depend on $\varphi$ through $E_n(\varphi) = \Delta^{*}(T)f(\varphi)$, where the generic function $f(\varphi)$ is no longer anti-symmetric with respect to $\varphi$, i.e., $f(-\varphi) \neq f(\varphi)$, reflecting the broken space-inversion and time-reversal symmetries, and the therefrom resulting nontrivial $\varphi_0$-phase shifts. Note that the only impact of temperature on $E_n(\varphi)$ is an effective rescaling (i.e., suppression with increasing temperature) of the superconducting-gap amplitude $\Delta^{*}(T)$ according to $\Delta^{*}(T) = \Delta^{*}(0)\tanh(1.74\sqrt{T_c/T - 1})$, whereas the qualitative shape of $E_n(\varphi)$ is not altered by temperature. The total Josephson current is then given by

$$I(\varphi) = \sum_n I(E_n(\varphi); T=0)\tanh\left(\frac{E_n(\varphi)}{2k_B T}\right), \qquad (5)$$

where the sum over $n$ ensures to account for the current contributions of all bound states (i.e., from all transverse channels of the junction) and $k_B$ indicates the Boltzmann constant. The current at zero temperature can, in the simplest case, be extracted from the thermodynamic relation[74]

$$I(E(\varphi); T=0) = -\frac{e}{\hbar}\frac{\partial E(\varphi)}{\partial \varphi} \qquad (6)$$

where $E = \sum_n E_n$ and $e$ is the positive elementary charge. The major temperature effect on the Josephson current originates, therefore, from the suppression of the higher-harmonic contributions in Eq. (6) due to the tanh term in Eq. (5).

The strengths of the Rashba SOI, the scalar (barrier), and the Zeeman potentials are measured by the dimensionless parameters $\lambda_{SOI} = m\alpha_R/(\hbar^2 k_F)$, $Z = 2mV_0d/(\hbar^2 k_F)$, and $\lambda_Z = 2mV_Zd/(\hbar^2 k_F)$, respectively, where $k_F = \sqrt{2m\mu}/\hbar$ refers to the Fermi wave vector (for the experimental parameters, $k_F \approx 3 \times 10^8\,m^{-1}$). In agreement with our earlier studies[27,60], we assume $Z = 0.5$−mimicking an average junction transparency of $\bar{\tau} = 1/[1 + (Z/2)^2] \approx 0.94$−and $\lambda_{SOI} = 0.661$−corresponding to Rashba SOI $\alpha_R \approx 15\,meV\,nm$. For a typical $g$-factor of $|g^{*}| \approx 12$, the Zeeman parameter $\lambda_Z = -0.40$ used for our theoretical calculations in the main text corresponds to the magnetic field $B_y \approx -100\,mT$.

## Data availability

The data that support the findings of this study are available at the online depository EPUB of the University of Regensburg, with the identifier DOI: 10.5283/epub.57878.

## Code availability

The computer codes that support the theoretical results, the plots within this paper and other findings of this study are available from the corresponding author upon request.

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

## Acknowledgements

We thank Max Geier, Abhishek Banerjee, and Karsten Flensberg for fruitful discussions. Work at Regensburg University was funded by the by EU's HORIZON-RIA Programme under Grant No. 101135240 (JOGATE), by the Deutsche Forschungsgemeinschaft (DFG, German Research Foundation) through Project-ID 314695032—SFB 1277 (Subprojects B05, B07, and B08)—and Project-ID 454646522—Research grant "Spin and magnetic properties of superconducting tunnel junctions" (A.C. and J.F.). D.K. acknowledges partial support from the project IM-2021-26 (SUPERSPIN) funded by the Slovak Academy of Sciences via the programme IMPULZ 2021 and Grant DESCOM VEGA 2/0183/21.

## Author contributions

S.R., T.A. and J.B. fabricated the devices and performed initial transport characterization of the hybrid superconductor/semiconductor wafer. S.R. and T.A. performed the measurements with the SQUID device. S.R., C.S. and N.P. conceived the experiment. A.C., D.K. and J.F. formulated the theoretical model. A.C. performed the numerical simulations of the temperature dependence of the SDE and $\varphi_0$. N.P., S.R., C.S., A.C. and D.K. wrote the manuscript. T.L., S.G. and G.C.G. designed the heterostructure and conducted MBE growth. S.R., T.A. and N.P. analyzed the data. M.J.M. supervised research activities at Purdue.

## Funding

## Competing interests

The authors declare no competing interests.
