## [Peer Review File · Nature Communications]

REVIEWER COMMENTS

Reviewer #1 (Remarks to the Author):

The authors study the relationship between the anomalous CPR and the supercurrent diode effect in an Al/InAs Josephson junction embedded in an asymmetric SQUID. The SQUID is needed for the phase tuning of the Josephson junction but inevitably complicates the identification of the CPR and the characterization of the non-trivial supercurrent diode effect. The authors considered various spurious phase shifts comprehensively and managed to quantitatively extract the anomalous phase shift of the CPR and the strength of the diode effect from the DC critical current of the SQUID in a reasonable way. The authors find that the supercurrent diode effect can be absent in large parallel magnetic fields and high temperatures where the anomalous phase shift is still observed. I understand that extracting such information in the SQUID devices can be quite challenging. However, I would like to ask a few questions to make sure that the analysis here is robust.

The method of extracting ϕ_0 is based on the model presented in Supplementary XI. While Fig. S2d nicely matches with Fig. S2c on a zoomed-out view, the experimental data presented in Fig. 1e clearly differs from Fig. S2b. In the experiment, as the gate voltage increases, the curve clearly shifts up while the simulated curve only has its oscillation amplitude increasing. What is the reason here? Note that the overall up-shifting can affect the intersection point with the baseline curve and thus affect the extracted shift of ϕ_0 .

The model nicely takes into account the inductive phase shift and Fraunhofer phase shift of JJ2 but does not consider such effects for JJ1. These phase shifts can affect the accuracy of extracting ϕ_0 and the diode efficiency. Can the authors give an estimate of how small these effects are based on comparison to the inductance and geometry of JJ2?

The curves of anomalous phase shift vs. V_g presented in Fig. 1g are obtained around $B_z = 14$ uT. Is the result robust against B_z ? Can the authors provide additional figures of Fig. 1g extracted at various B_z in the supplementary?

The authors claim that ϕ_0 shift "is a necessary but not sufficient condition for the SDE". The data presented in Fig. 2f & Fig. 3 supports the argument that ϕ_0 shift is not sufficient for SDE. However, it does not mean that ϕ_0 shift is necessary. Have the authors examined devices with minimal SOI such as graphene?

The theory shows that SDE dies exponentially with increasing temperature (Fig. 3b) while the experiment only shows a linear relationship. Can the authors explain the discrepancy here?

The authors indicate that the monotonic relation between the phase shift and gate voltage is due to the proportionality between the Rashba SOI strength and the electron density. However, in Ref 61, this proportionality is only observed in a short range of electron density (0.7 to $1.7 \cdot 10^{-12}$ cm⁻²). What is the corresponding electron density of the device studied here in the relevant range of V_g (-1 to 1 V)?

Additional comments:

The model presented in the supplementary is necessary for understanding the data analysis. I recommend moving it (and the associated reasonings) to the main text.

In the top right corner of page-3, the authors writes "the graph clearly shows the proportionality of $\Delta\phi$ to $-e_z (B_{ip} \times I)$ ". I recommend the authors refer to Fig. S4b here.

Reviewer #2 (Remarks to the Author):

Authors study the gate- and field-dependence of the anomalous phase and diode effect in the same junction relative to a large reference junction in an asymmetric SQUID. The anomalous phase is extracted from the crossing of the SQUID oscillations with a reference curve obtained with the target junction pinched off; the diode effect is extracted from the amplitude of the SQUID oscillations relative to this reference curve. Overall the message of the paper is clear, and the gate-dependence in particular seems to rule out a purely diamagnetic/orbital mechanism. My concerns and comments are below:

- In Fig. 1 and elsewhere in main text, be consistent with $\theta=-90\text{deg}$ vs. $\theta=270\text{deg}$, so it's easier to identify identical configurations

- in Fig. 2e why the diode effect should be 0 at all B...is this incidental esp near $V_g=0$? there is no B dependence?

- Why is the range of gate voltage limited to $[-1.5\text{ V}, 1.0\text{ V}]$? how much density is changing? Presumably Fig. S1 is a Hall bar device. But even there no "n" vs V_g is presented.

- On p. 3: "Data in Fig. 2e(i-v) show that $d\eta/dV_g$ is linear in B_y " ...I don't see this...I do see that η^* is linear in V_g^* (for the range of voltage shown), and *perhaps* one could argue that Fig. *2f* shows $d\eta/dV_g$ is linear in B_y at fields $\ll 50\text{mT}$ (although there are only a few points)...this is clarified a few sentences later, but I was lost at the initial sentence

- The model in the supplement used to justify the method of extracting both $\Delta\phi$ from the crossings with the reference curve and $I_{c\pm}$ from the amplitude above/below the reference curves uses a sinusoidal CPR, but the importance of higher harmonics for the SDE is stressed towards the end of the main text. What are the implications of a skewed CPR for your methodology of extracting $\Delta\phi$ and $I_{c\pm}$ of the target junction from the SQUID oscillations?

- In the supplement, it is stated "one can still deduce I_{c2+} from the maximum of the oscillations, since these two quantities are proportional." since I_{max} is not proportional to I_{c2+} this needs clarification

- In Fig. 3, $\Delta\phi$ is measured at 100mT where diode effect is suppressed (see Fig. 2f), while diode effect is measured at 50mT where diode effect is maximal...is this a fair comparison? Why do "screening effects" make anomalous phase measurement hard at smaller field?

Reviewer #3 (Remarks to the Author):

The authors report on a detailed comparison between the φ_0 shift and the superconducting diode effect (SDE). This is a difficult task due to constrain in measuring the Josephson current phase relation and the critical current asymmetry in the same device. A key ingredient of their work is the gate voltage tunability of their weak link which is used to tune the φ_0 and the SDE. Moreover, striking additional experiments are the temperature dependences of the SDE and φ_0 effects. Their findings, compared with theoretical simulation of the device geometry, screening effects, Zeeman field and Rashba spin-orbit coupling parameters, deepen our understanding of these new superconducting phenomena. The study undoubtedly offers valuable insights for a broad audience and especially for those who focus on 2D materials and superconducting quantum devices. Consequently, I believe this study would be a fitting contribution to Nature Communication. However, I recommend addressing a few points to get better focus on the main findings of their work and make the interpretation of the data more comprehensive.

In particular, the link between the intrinsic SDE and the φ_0 is not as obvious as the authors claim and did not clearly emerge to me from the data. Therefore, I ask the authors to answer the following questions and to modify the manuscript accordingly to make it more comprehensive. If these questions can not be convincingly answered, the author should substantially reduce their claim about the «evident» link between intrinsic SDE and φ_0 .

1) While the SDE is obvious from the data (already in Fig. 1c), the authors try to isolate the intrinsic (or SOI based) SDE which is small in this device (SDE efficiency <10%). This is complicated because it requires to cancel most of the SDE already present in the device due to screening effect. This is accomplished by comparing $I_{c+}(B_z)$ and $I_{c-}(-B_z)$, instead of I_{c+} and I_{c-} at the same B_z . This is reasonable, but it should be written in the main text SDE discussion that this analysis is used to cancel the SDE due to screening effects.

2) A crucial statement made by the author to compare the φ_0 and the intrinsic SDE is that the intrinsic SDE is proportional to $d\eta/dV_g$. The arguments which justify this procedure are given in the supplementary but they should appear in the main text as it is central to the data interpretation.

3) I think that the claimed link between intrinsic SDE and φ_0 is made too strongly by the author and not supported by the data. «The results shown so far provide strong evidence of the link between anomalous phase shift and SDE. Both effects are linear in B_y .» It should be written that this is only true at low in plane field for the SDE. (As a side note, the supplementary should be cited in the main text for the linear B_y dependence of φ_0 .) Moreover, the authors' theory does not explain why the SDE vanishes at such a low in plane magnetic field, can the author explain this with their theory model ?

4) In order to better separate the orbital mechanism and the SOI based SDE, is it possible to see Fig. 2f at higher in plane field ? I expect that the SDE parameter η will stick to zero at higher in plane field if the author interpretation of multiple harmonic and SOI is correct. However, the SDE might be recover at higher in plane field for the orbital mechanism.

5) Regarding these statements : « Both $\Delta\varphi_0$ and η can be modulated by a gate voltage in the accessible range $V_g \in [-1.5 \text{ V}, 1.0 \text{ V}]$ » and after « The relation between η and φ_0 is evident from the comparison between Fig. 1g and Fig. 2e. » While I do agree that both effects are due to intrinsic

properties of the barrier since they are modulated by the gate voltage, I do not agree that the SDE can be strictly attributed to the combination of Zeeman and SOI like the φ_0 . The φ_0 increases with gate voltage, this is not the case of $d\eta/dVg$ which is assumed to be proportional to η according to the author. Also, I could imagine that the gate voltage depletes the carrier in the N material, leading to a reduced thickness of the proximitized region and to a variation of the SDE based on the orbital mechanism. Can the authors comment on that ?

6) The temperature dependence is maybe the most striking evidence for different mechanism between the SDE and the φ_0 . While the authors show that the SDE necessitate, in addition to the φ_0 ingredients, multiple Andreev level participating in transport, this statement is not enough backed up by data to be convincingly made. If, as the authors argue, the SDE is lost because the current contribution of low energy Andreev levels, each with a given φ_0 , is lost ; then why does the φ_0 is constant as a function of temperature ? I would expect that the mean measured φ_0 also change as a function of temperature based on the author interpretation. But this not the case experimentally, as the authors show. Can the authors comment on that ?

ANSWERS TO THE QUESTIONS BY THE REFEREES

Referee 1

Reviewer #1 (Remarks to the Author): The authors study the relationship between the anomalous CPR and the supercurrent diode effect in an Al/InAs Josephson junction embedded in an asymmetric SQUID. The SQUID is needed for the phase tuning of the Josephson junction but inevitably complicates the identification of the CPR and the characterization of the non-trivial supercurrent diode effect. The authors considered various spurious phase shifts comprehensively and managed to quantitatively extract the anomalous phase shift of the CPR and the strength of the diode effect from the DC critical current of the SQUID in a reasonable way. The authors find that the supercurrent diode effect can be absent in large parallel magnetic fields and high temperatures where the anomalous phase shift is still observed. I understand that extracting such information in the SQUID devices can be quite challenging. However, I would like to ask a few questions to make sure that the analysis here is robust.

We thank the Referee for the positive feedback and for the attentive reading of our manuscript. In what follows, we provided a detailed answer to each question.

1) *The method of extracting φ_0 is based on the model presented in Supplementary XI. While Fig. S2d nicely matches with Fig. S2c on a zoomed-out view, the experimental data presented in Fig. 1e clearly differs from Fig. S2b. In the experiment, as the gate voltage increases, the curve clearly shifts up while the simulated curve only has its oscillation amplitude increasing. What is the reason here? Note that the overall up-shifting can affect the intersection point with the baseline curve and thus affect the extracted shift of φ_0 .*

We want to stress that also the theoretical curves “shift upwards”, this is in the new supplementary made more visible since we added two more curves with higher critical current (500 nA and 600 nA) to Fig. S2b.

To be more precise, this is not, strictly speaking a real shift upwards, it is a visual effect due to the fact that the negative side of the oscillation gets “cut” at higher values. We refer here to the new Fig.S2b. Notice that the curves all cross at the same point with the zero critical current curve. The important point is that the qualitative behavior of the simulated curves is the same as the experimental ones. Therefore, our method of extracting φ_0 is not affected by the shift.

2) *The model nicely takes into account the inductive phase shift and Fraunhofer phase shift of JJ2 but does not consider such effects for JJ1. These phase shifts can affect the accuracy of extracting φ_0 and the diode efficiency. Can the authors give an estimate of how small these effects are based on comparison to the inductance and geometry of JJ2?*

Indeed, we do consider these effects for JJ1 –actually we consider for the Fraunhofer pattern *only* JJ1– and we neglect instead them for JJ2.

We motivate here why we neglect the effects for JJ2 (in case the Referee meant JJ2 instead of JJ1). For the small junction (JJ2) the periodicity of the Fraunhofer interference pattern will be given by the field $B_z = \Phi_0/(aw)$, with the width $w = 2.7 \mu\text{m}$ and the effective length $a \sim 1 \mu\text{m}$ (this value was found experimentally in Ref. 60). From this we obtain a periodicity of 800 μT . As the applied fields $|B_z| < 20 \mu\text{T}$ are much smaller we can safely ignore interference effects in JJ2.

3) *The curves of anomalous phase shift vs. V_g presented in Fig. 1g are obtained around $B_z = 14 \mu\text{T}$. Is the result robust against B_z ? Can the authors provide additional figures of Fig. 1g extracted at various B_z in the supplementary?*

We have extracted the anomalous phase shift from different SQUID oscillations (i.e., at different B_z) and the results are nearly equal. To show this, we have added a new figure to the supplement. In Figure S5a,b we now show the evaluation of $\Delta\varphi_0$ from three different SQUID oscillations. We find that our results do not depend on the chosen oscillation interval.

As an additional consistency check we have also extracted the anomalous phase shift from measurements at negative current bias. The resulting data is shown in Figure S5c,d. The obtained $\Delta\varphi_0$ is in good agreement with the data obtained at positive current bias.

4) *The authors claim that φ_0 shift “is a necessary but not sufficient condition for the SDE”. The data presented in Fig. 2f & Fig. 3 supports the argument that φ_0 shift is not sufficient for SDE. However, it does not mean that φ_0 shift is necessary. Have the authors examined devices with minimal SOI such as graphene?*

This is an important question, which is about the main message of our work.

We acknowledge that our statement reported by the Referee was too strong. A rigorous and conclusive experimental demonstration of the necessity of φ_0 for the observation of supercurrent rectification is perhaps impossible in a single set of experiments, since this would always leave open the possibility of the observation of η without φ_0 in another material system or in different conditions. Nevertheless, in our work we present at least strong indications that the observation of φ_0 and η are correlated in the way expected by the simplest model based on spin-orbit effects. In particular, what we do in our work is:

- (i) first, we demonstrate a device where the measurement of η and φ_0 is simultaneously possible;
- (ii) we show that in ballistic Josephson junctions at the lowest temperature regime, the observation of anomalous Josephson shift φ_0 supercurrent rectification η are correlated, in the sense that they show a similar dependence versus external parameters as gate voltage or in-plane field .
- (iii) In some conditions (above a certain in-plane field threshold, or above 600 mK), the φ_0 shift is observed, while the supercurrent rectification is, instead, suppressed.

Both (ii) and (iii) agree with predictions of the simplest model based on spin-orbit interaction effects in Josephson junctions. In the *Discussion* section, we state that our experiment does not exclude that an experimental realization of SDE might exist, which is not accompanied by φ_0 shift. What we do is to show that in ballistic JJs with Rashba spin-orbit interaction, SDE is always accompanied by φ_0 , and both respond to the variation of external parameters in the same way as expected by spin-orbit based models.

Our work is important because it allows to conclude that alternative models based on orbital effects (Ref. 30 and 50) are at least not able to explain an important part of the observations (i.e., the gate voltage dependence of both φ_0 and η), and thus that the SOI mechanism seems necessary to explain the supercurrent rectification.

In her/his last question the Referee asks about possible measurements in materials with low spin-orbit interaction. This would certainly be interesting. At the moment, our group is not working on Josephson junctions in materials with low spin-orbit interaction. However, we might perhaps comment on the fact that, despite the significant amount of experiments on superconducting diodes appeared in the literature in recent times, we are not aware of experiments reporting intrinsic SDE in materials with low spin-orbit interaction (SDE in magic angle graphene has been reported, but in films and not in SNS junctions –the SDE mechanism is in bilayer graphene completely different).

Finally, we would like to point out that the absence of spin-orbit interaction does not mean, in principle, absence of φ_0 . For example, the theoretical mechanism in Ref.30 and 50 does not require spin-orbit interaction, and *still it shows both φ_0 and η* . However, there the gate dependence of φ_0 is absent.

We acknowledge that our statement in the intro about the φ_0 -shift being a necessary condition for the SDE was too strong. We have therefore corrected it.

We changed a sentence in the Intro: “we demonstrate that the former is a necessary but not sufficient condition for the SDE. This latter requires in fact the presence of higher harmonics in the CPR, which are quickly suppressed by increasing the temperature.” This now reads “**we highlight the role of the higher harmonics of the CPR for the emergence of the SDE**”.

At the end of the discussion of Fig. 2 we exchanged the sentence: “The φ_0 -shift is a necessary but not sufficient condition for the supercurrent nonreciprocity.” with the sentence “**However, the φ_0 -shift is, *per se*, not a sufficient condition for the supercurrent nonreciprocity.**”

5) *The theory shows that SDE dies exponentially with increasing temperature (Fig. 3b) while the experiment only shows a linear relationship. Can the authors explain the discrepancy here?*

As we show in the supplemental Fig.S6c,d the error in the determination of the SDE increases strongly at higher temperatures. This is because of the decrease of critical current at higher temperatures and a larger error in the determination of I_c from the $V(I)$ curves. From the present data we can conclude that the SDE decreases strongly with temperature, but we do not have the resolution at small I_c to see an exponential decrease of $d\eta/dV_g(T)$.

6) *The authors indicate that the monotonic relation between the phase shift and gate voltage is due to the proportionality between the Rashba SOI strength and the electron density. However, in Ref 61, this proportionality is only observed in a short range of electron density (0.7 to $1.7 \cdot 10^{12} \text{ cm}^{-2}$). What is the corresponding electron density of the device studied here in the relevant range of V_g (-1 to 1 V)?*

Finding the actual electron density in the 2DEG is difficult owing to the electrostatic screening of the Al layer. We can find the density in Hall-bar devices where the aluminum is removed, as done in the supplement (and in our previous works). The density found from Hall-bar samples cannot be directly used to estimate that in junctions, as there the presence of the aluminum electrodes largely screens the effect of the top gate. This difficulty is well known in the literature, see e.g. our previous work in Ref.60 and supplement therein.

In Hall bar measurements with Al stripped, see Fig.S1 of the present manuscript, the density is of the order of 10^{12} cm^{-2} .

7) *Additional comments: The model presented in the supplementary is necessary for understanding the data analysis. I recommend moving it (and the associated reasonings) to the main text.*

We agree with the referee that the findings from the simulations are important for the data analysis and should be more visible in the paper. We now introduce the model in the Methods section of the main text. We left the detailed description of the model and the simulation results in the supplement, as we believe that they would be too lengthy for the main text.

8) *In the top right corner of page-3, the authors writes “the graph clearly shows the proportionality of $\Delta\varphi_0$ to $-e_z(B_{ip} \times I)$ ”. I recommend the authors refer to Fig. S4b here.*

We have added a link to the text there, stating that the sinusoidal dependence of $\Delta\varphi_0(\theta)$ is shown in Fig. S4. The added text is: “**A graph of $\Delta\varphi_0$ versus the angle θ is shown in Fig. S4 of the Supplemental Information. As expected, we find the sinusoidal dependence $\Delta\varphi_0 \sim -\sin(\theta)$.**”

Referee 2

Reviewer #2 (Remarks to the Author): Authors study the gate- and field-dependence of the anomalous phase and diode effect in the same junction relative to a large reference junction in an asymmetric SQUID. The anomalous phase is extracted from the crossing of the SQUID oscillations with a reference curve obtained with the target junction pinched off; the diode effect is

extracted from the amplitude of the SQUID oscillations relative to this reference curve. Overall the message of the paper is clear, and the gate-dependence in particular seems to rule out a purely diamagnetic/orbital mechanism. My concerns and comments are below:

We thank the Referee for highlighting the crucial aspects of our work and for her/his constructive questions.

1) - In Fig. 1 and elsewhere in main text, be consistent with $\vartheta = -90\text{deg}$ vs. $\vartheta = 270\text{deg}$, so it's easier to identify identical configurations

We changed the labels in Fig. 1g and now consistently use angles from -180 to 180 degrees everywhere in the text.

2) - in Fig. 2e why the diode effect should be 0 at all B_y ...is this incidental esp near $V_g = 0$? there is no B dependence?

Due to SQUID screening effects, our measurements of $\eta(V_g)$ are affected by an unknown vertical offset which depends on the magnitude and direction of the in-plane field. This is why we focus only on the slope, i.e., $d\eta/dV_g$. We believe that the zero-crossings of η at $V_g = 0$ are mostly coincidental; also, not all the subpanels of Fig.2e have $\eta(B_y = 0) = 0$.

3) - Why is the range of gate voltage limited to [-1.5 V, 1.0 V]? how much density is changing? Presumably Fig. S1 is a Hall bar device. But even there no " n " vs V_g is presented.

When going above 1 V with the gate voltage, we observe the onset of a leakage current flowing into the gate. Therefore, we are unable to use higher gate voltages. When going below -1.5 V, the critical current of the junction becomes very low and evaluating $\Delta\varphi_0$ and the diode effect becomes difficult. Regarding the density in the junctions, we refer to question 6 of Referee #1. We do not show the gate voltage dependence of n in Fig.1S, as this data from a Hall bar device cannot be applied to the junction devices.

4) - On p. 3: "Data in Fig. 2e(i-v) show that dr/dV_g is linear in B_y "...I don't see this...I do see that η is linear in V_g (for the range of voltage shown), and perhaps one could argue that Fig. 2f shows dr/dV_g is linear in B_y at fields $\ll 50\text{mT}$ (although there are only a few points). . . this is clarified a few sentences later, but I was lost at the initial sentence

The linearity of η (and thus of $\partial\eta/\partial V_g$, which is easier to measure for us) as a function of $\vec{B}_{ip} \times \vec{l}$ is valid only up to a certain threshold value of the in-plane field. After that threshold, most experiments (ours and from the literature, see Refs. 18, 22-28) on Josephson diodes show an abrupt suppression of the rectification. The fact that such threshold field is relatively small (depairing is still not relevant at such fields, see e.g. Fig.S6 in Ref.18) together with the sharp discontinuity in the experimental $\partial\eta/\partial B_{ip}$ is an intriguing observation which motivated our previous work in Ref. 27. There, we measured the slope discontinuity for field dependence of both η and the magnetochiral anisotropy for the inductance (two experimentally independent observations) and we found that the outcome of the experiments perfectly match the analytical prediction of a minimal model. From this we deduced that the discontinuity in η (and possibly its sign change) is due to a so-called $0-\pi$ -like transition, first introduced in Ref.14. Such transition is expected to occur in systems featuring many channels, spin-orbit interaction and Zeeman field. It is important to add that our model does not predict any discontinuity in φ_0 ; however, our experiment in Ref.27 was not able to measure φ_0 .

In our opinion, all this was discussed in detail in our previous work (Ref.27) and it is beyond the scope of the present manuscript. Here, our goal is instead to experimentally show the tight relation between φ_0 (now finally accessible to the experiment) and η . These two quantities have a similar behavior as long as B_y is below the $0-\pi$ -like transition.

We modified the text to make it more clear and self-contained (in the previous version we simply referred to our previous work in Ref.27). It is clear, however, that an exhaustive discussion of the $0-\pi$ -like transition is beyond the scope of this manuscript, and that we must thus necessarily refer to the Ref.27 for details.

We added the following sentence (part of a more comprehensive modification of the text described in the answer to Question 5 of Referee 3): "The suppression at higher field was interpreted in Ref. [27] as a signature of a $0-\pi$ -like transition, occurring when two minima in the Josephson energy-phase relation become degenerate."

5) - The model in the supplement used to justify the method of extracting both $\Delta\varphi$ from the crossings with the reference curve and $I_{c\pm}$ from the amplitude above/below the reference curves uses a sinusoidal CPR, but the importance of higher harmonics for the SDE is stressed towards the end of the main text. What are the implications of a skewed CPR for your methodology of extracting $\Delta\varphi$ and $I_{c\pm}$ of the target junction from the SQUID oscillations?

Before answering the question, we would like to stress that (again due to SQUID screening) oscillations in $I_c(B_z)$ do not reproduce exactly the shape of the current phase relation. In fact, a SQUID with screening always shows a skewed $I_c(B_z)$ even if the CPR is sinusoidal. In our case, for the arguments provided in the text, we claim that the CPR is indeed skewed, but this skewness is not intrinsic, but caused by screening.

That said, we come to the Referee's question. The extraction of φ_0 with our technique does not depend on the shape of the CPR. We find the point where $I_{c,2}(\varphi) = 0$ which is independent of the shape of the junction's CPR. To confirm explicitly this, we have done an additional simulation with our circuit model where we now use a fully ballistic CPR for the probe junction (JJ2). The resulting $I_c(B_z)$ curves are shown in the new Fig. S2g. With the ballistic CPR we also obtain the proportionality between the extracted critical current and the actual critical current of the junction, as shown in Fig. S2h.

6) - In the supplement, it is stated "one can still deduce I_{c2+} from the maximum of the oscillations, since these two quantities are proportional." since I_{max} is not proportional to I_{c2+} this needs clarification

In order to make this statement in the supplement more clear, we have added Fig.S2e,f, where we use the simulated critical current to explain the extraction of $I_{c,2}$. Fig.S2f shows the proportionality between the extracted critical current and the real critical current of the small junction (JJ2).

7) - In Fig. 3, $\Delta\varphi$ is measured at 100mT where diode effect is suppressed (see Fig. 2f), while diode effect is measured at 50mT where diode effect is maximal...is this a fair comparison? Why do "screening effects" make anomalous phase measurement hard at smaller field?

This is discussed in the supplementary section "MODELING OF INDUCTIVE SCREENING EFFECTS IN THE ASYMMETRIC SQUID", see the last bullet point. The key point is that the screening effect depends on the product of critical current times inductance (in the Tinkham's book this is called screening parameter $\beta_m \equiv 2LI_c/\Phi_0$). By increasing the magnetic field (at least in the field regime under study, which are well below the Al in-plane critical field), the critical current decreases much faster than L increases, thus the β_m parameter (and thus the screening effects) become *less* important at higher fields.

As can be seen in Fig. 2 **b,c** ($B_{ip} = 50$ mT) the SQUID oscillations no longer intersect with the baseline ($V_g = -2.5$ V), therefore we cannot use our technique of extracting φ_0 for fields lower than 100 mT. This is now stated more explicitly in the supplement. We expect that $\varphi_0(B_y)$ has a linear behavior in the range $[-100$ mT, 100 mT]. This is supported by Fig.S4 where we study $\varphi_0(B_y)$ by rotating an in-plane field with constant magnitude $B_{ip} = 100$ mT.

Referee 3

Reviewer #3 (Remarks to the Author): The authors report on a detailed comparison between the φ_0 shift and the superconducting diode effect (SDE). This is a difficult task due to constrain in measuring the Josephson current phase relation and the critical current asymmetry in the same device. A key ingredient of their work is the gate voltage tunability of their weak link which is used to tune the φ_0 and the SDE. Moreover, striking additional experiments are the temperature dependences of the SDE and φ_0 effects. Their findings, compared with theoretical simulation of the device geometry, screening effects, Zeeman field and Rashba spin-orbit coupling parameters, deepen our understanding of these new superconducting phenomena. The study undoubtedly offers valuable insights for a broad audience and especially for those who focus on 2D materials and superconducting quantum devices. Consequently, I believe this study would be a fitting contribution to Nature Communication. However, I recommend addressing a few points to get better focus on the main findings of their work and make the interpretation of the data more comprehensive. In particular, the link between the intrinsic SDE and the φ_0 is not as obvious as the authors claim and did not clearly emerge to me from the data. Therefore, I ask the authors to answer the following questions and to modify the manuscript accordingly to make it more comprehensive. If these questions can not be convincingly answered, the author should substantially reduce their claim about the «evident» link between intrinsic SDE and φ_0 .

We thank the Referee for the positive comments. The Referee asked important questions which helped us to better clarify the key points of our manuscript.

1) While the SDE is obvious from the data (already in Fig. 1c), the authors try to isolate the intrinsic (or SOI based) SDE which is small in this device (SDE efficiency <10%). This is complicated because it requires to cancel most of the SDE already present in the device due to screening effect. This is accomplished by comparing $I_c+(B_z)$ and $I_c(-B_z)$, instead of I_c+ and I_c- at the same B_z . This is reasonable, but it should be written in the main text SDE discussion that this analysis is used to cancel the SDE due to screening effects.

We add the following sentence in the main text when presenting Fig1c: "The dominant extrinsic SDE of the SQUID as a whole (since $I_c(B_z) \neq -I_c(-B_z)$) is a long-known consequence of screening in the loop, which must be kept distinct from the intrinsic SDE in single homogeneous junctions, which is the effect we are studying. As discussed in the Supplementary Information, screening in SQUIDs arises when there are current-dependent corrections to the fundamental SQUID relation"

Later on, we add the following: "We stress that, owing to screening, one cannot extract the negative critical current in the CPR from the relative minima of $I_c^+(B_z)$. As discussed in the Supplementary Information, it is instead necessary to look at the maxima of $|I_c^-(B_z)|$."

2) A crucial statement made by the author to compare the φ_0 and the intrinsic SDE is that the intrinsic SDE is proportional to $d\eta/dV_g$. The arguments which justify this procedure are given in the supplementary but they should appear in the main text as it is central to the data interpretation.

We agree that the circuit model and the conclusions from the simulations are important for the understanding of the data interpretation. We now introduce the circuit model in the Methods section of the main text and discuss the most important conclusions from the simulations. For the technical details of the model and the simulation results we refer to the supplement.

Concerning the proportionality between η and $\partial\eta/\partial V_g$, we refer to the answer 5 below.

3) I think that the claimed link between intrinsic SDE and φ_0 is made too strongly by the author and not supported by the data. «The results shown so far provide strong evidence of the link between anomalous phase shift and SDE. Both effects are linear in B_y .» It should be written that this is only true at low in plane field for the SDE. (As a side note, the supplementary should be cited in the main text for the linear B_y dependence of φ_0 .) Moreover, the authors' theory does not explain why the SDE vanishes at such a low in plane magnetic field, can the author explain this with their theory model ?

For what concerns the too strong statement about the link between φ_0 and η , we refer to our answer to the question 4 of Referee 1, and text modifications indicated therein.

For what concerns the relation between $\partial\eta/\partial V_g$ and B_y , we have modified the text as described in the answer to question 4 of Referee #2. Regarding the linear B_y dependence of $\Delta\varphi_0$ we have added a link to the supplemental Fig.S4.

4) *In order to better separate the orbital mechanism and the SOI based SDE, is it possible to see Fig. 2f at higher in plane field ? I expect that the SDE parameter η will stick to zero at higher in plane field if the author interpretation of multiple harmonic and SOI is correct. However, the SDE might be recover at higher in plane field for the orbital mechanism.*

We do not have measurements of the SDE parameter for fields higher than 100 mT. In our previous work Ref. 27 we have measured η for single junction devices for in-plane fields up to 500 mT. We did not observe a recovery of the diode effect up to this field.

5) *Regarding these statements : « Both $\Delta\varphi_0$ and η can be modulated by a gate voltage in the accessible range $V_g \in [-1.5 V, 1.0 V]$ » and after « The relation between η and φ_0 is evident from the comparison between Fig. 1g and Fig. 2e. » While I do agree that both effects are due to intrinsic properties of the barrier since they are modulated by the gate voltage, I do not agree that the SDE can be strictly attributed to the combination of Zeeman and SOI like the φ_0 . The φ_0 increases with gate voltage, this is not the case of $d\eta/dV_g$ which is assume to be proportional to η according to the author. Also, I could imagine that the gate voltage depletes the carrier in the N material, leading to a reduced thickness of the proximitized region and to a variation of the SDE based on the orbital mechanism. Can the authors comment on that ?*

We start commenting the first part of this question, which is partially related to the question 4 of Referee 1. We agree with the Referee that, in the first version of the manuscript, in some points of the main text our statements were too strong, and we edited them in the new version, see below. However, we would like to stress that in the Discussion section we wrote (and still write in the new version) that “The observed strong gate dependence of φ_0 and η indicates that the Rashba-based mechanism must certainly play an important role in the SDE. On the other hand, the orbital mechanism cannot be ruled out by our observations: it could still coexist with the spin-orbit-based mechanism”.

The Referee remarks that, while φ_0 increases with V_g , $\partial\eta/\partial V_g$ does not. We remark that a change in the V_g dependence of $\partial\eta/\partial V_g$ means a change in the curvature of $\eta(V_g)$. In a measurement like that in Fig.2e, the scatter of the data points and their error bars do not allow one to make conclusive statements on the curvature of $\eta(V_g)$. This is due to the objective difficulties in measuring the diode effect in SQUIDS with large kinetic inductance (and thus screening effects). Despite these difficulties (which were carefully taken into account) our data clearly show that: (i) both $\partial\eta/\partial V_g$ and $\partial\varphi_0/\partial V_g$ monotonically increase with B_y , which is at least a strong indication of the same behavior for η and φ_0 ; (ii) both $\partial\eta/\partial V_g$ and $\partial\varphi_0/\partial V_g$ are zero if the in-plane field is directed along \hat{x} : this is a strong indication of the fact that the same is true for η and φ_0 ; (iii) η and φ_0 change substantially when the gate voltage is changed in the same range. The latter is an important observation, because orbital effects are expected to depend very little on the gate voltage. This brings us to the second part of the Referee’s comment.

Owing to the electrostatic screening of the Al layer placed just 10 nm above the 2DEG, a change in the gate voltage can have at most an effect only on a region of few tens of nm next to the Al gap at the weak link. Since our leads are large (the length is many micrometers) one would expect a negligible effect of the gate voltage on the orbital mechanism. The fact that we do see an effect is at least a clear indication of the role of spin-orbit interaction. On the other hand, we admit (as we do in the Discussion) section that the orbital mechanism is possible as an *additional* mechanism: the two mechanisms may coexist.

Finally, we notice that the original statement in the main text “...the modulation of η with V_g is roughly proportional to η itself” is in a strict sense not correct (unless $\eta(V_g)$ is exponential), and misleading. In this paragraph, what we mean is that a change in V_g does not change the shape of $\eta(B_y)$ or $\eta(T)$, but it only rescales it (i.e., η can be factorized in a function of V_g times a function of \vec{B}, T etc.) If we assume so, then *seen as a function of \vec{B} or T* (or other any other parameter) η and $\partial\eta/\partial V_g$ are mutually proportional. Therefore we can use $\partial\eta/\partial V_g$ as a figure of merit for η when studying the field or temperature dependence.

We changed the main text as follows:

*As explained in the Supplementary Information, η is subjected to a spurious vertical shift owing to SQUID screening effects . On the other hand, its slope as a function of V_g , i.e. $\partial\eta/\partial V_g$ remains unaffected by screening. Our crucial assumption is that, *seen as functions of \vec{B} or T* , η and $\partial\eta/\partial V_g$ are proportional, i.e., the larger η , the larger is its modulation of as a function of V_g . Thus, we shall use $\partial\eta/\partial V_g$ as a convenient figure of merit for the magnetochiral effect when considering field or temperature dependence.*

Previous reports on Josephson diodes [18, 22, 24, 25] showed that the rectification efficiency is linear in B_y , but only up to a certain threshold (here at $|B_y| = 50$ mT): after that, a clear suppression is observed. Our data in Fig. 2f, deduced from data in Fig.2e, show precisely the same behavior. The suppression at higher field was interpreted in Ref. [27] as a signature of a 0- π -like transition, occurring when two minima in the Josephson energy-phase relation become degenerate.

6) *The temperature dependence is maybe the most striking evidence for different mechanism between the SDE and the φ_0 . While the authors show that the SDE necessitate, in addition to the φ_0 ingredients, multiple Andreev level participating in transport, this statement is not enough backed up by data to be convincingly made. If, as the authors argue, the SDE is lost because the current contribution of low energy Andreev levels, each with a given φ_0 , is lost ; then why does the φ_0 is constant as a function of temperature ? I would expect that the mean measured φ_0 also change as a function of temperature based on the author interpretation. But this not the case experimentally, as the authors show. Can the authors comment on that ?*

In our work, we argue that the SDE requires multiple *transverse* channels (this means that the junction width is many times the Fermi wavelength). Instead, concerning the junction *length*, we assume for simplicity that the junction is in the short limit, i.e.,

there is only one longitudinal pair of ABS (in the experiment, the junction is actually in an intermediate regime between short and long junction limit). However, what matters for the SDE is the number of transverse channels, namely, the junction width.

In fact, to observe the SDE we require *simultaneously* all the following three ingredients:

(i) spin-orbit + magnetic field induce a φ_0 shift proportional to $v_{F,x}$, the x -component of the Fermi velocity. (ii) since in a wide junction, the transverse channels have different $v_{F,x}$ and thus different φ_0 , the total current phase relation (CPR) will be the sum of CPRs with different shift. (iii) since trigonometry states that the sum of sinusoidal functions is always a sinusoidal function (thus reciprocal), nonreciprocity requires that the single channel CPRs are skewed (i.e. contain higher harmonics).

It is the ingredient (iii) [and not (ii)] that is relevant for the temperature dependence. At higher temperatures (as long as $\Delta^*/k_B T \ll 1$ is not valid anymore, see e.g. Furusaki-Beenakker formula*), individual channel CPRs become sinusoidal. Physically this is due to the suppression of coherent higher-order Andreev reflections which determine the higher harmonics. For this reason, an increase of temperature affects η owing to the suppression of the higher harmonics, but not φ_0 which is relatively insensitive (both in the model and in the experiment) to T .

The picture we just illustrated is discussed in the main text in the paragraph starting with “The results shown so far...”

(*) A. Furusaki and M. Tsukada, Solid State Commun. 78, 299 (1991); C.W.J. Beenakker and H. van Houten, Phys. Rev. Lett. 66, 3056 (1991).

A. Other changes in the text

Beside the changes mentioned in the answers to the Referees, we have implemented the following minor changes:

- In the *Methods* section we introduce the label n for the n -th transverse channel, see modifications in red in the pdf file for the reviewers.
- We added in the Acknowledgments the project JOGATE.

REVIEWERS' COMMENTS

Reviewer #1 (Remarks to the Author):

The authors have reasonably cleared my concerns. I can recommend publishing.

Reviewer #3 (Remarks to the Author):

I appreciate that the authors have adjusted their claim regarding the link between φ_0 and the superconducting diode effect. The last sentence of the abstract should be amended accordingly, as it is not demonstrated that the superconducting diode effect "mainly" arises from spin-orbit interaction, but rather only partially as the orbital mechanism is also at play. After this correction has been made, I would consider the manuscript suitable for publication in Nature Communications.

List of changes.

-1. Following the indication of Referee 3 we changed the last line of the abstract.

The final line of the abstract was:

“Our findings show that the supercurrent diode effect mainly results from magnetochiral anisotropy induced by spin-orbit interaction in combination with a Zeeman field.”

Now it reads:

“Our findings show that spin-orbit interaction in combination with a Zeeman field plays an important role in determining the magnetochiral anisotropy and the supercurrent diode effect.”

Other small changes.

-2. The author Denis Kochan adds the following affiliation

Affiliation #8: Center of Quantum Frontiers of Research and Technology (QFort), National Cheng Kung University, Tainan 70101, Taiwan

-3. The following is added in the Acknowledgements (addition in red) “D.K. acknowledges partial support from the project IM-2021-26 (SUPERSPIN) funded by the Slovak Academy of Sciences via the programme IMPULZ 2021 and Grant DESCOM VEGA 2/0183/21.”